# Differential Diagnosis of Solid Pancreatic Lesions Using Detective Flow Imaging Endoscopic Ultrasonography

**DOI:** 10.3390/diagnostics14090882

**Published:** 2024-04-24

**Authors:** Haruo Miwa, Kazuya Sugimori, Shoichiro Yonei, Hayato Yoshimura, Kazuki Endo, Ritsuko Oishi, Akihiro Funaoka, Hiromi Tsuchiya, Takashi Kaneko, Kazushi Numata, Shin Maeda

**Affiliations:** 1Gastroenterological Center, Yokohama City University Medical Center, Yokohama 232-0024, Japankz_numa@yokohama-cu.ac.jp (K.N.); 2Department of Gastroenterology, Yokohama City University, Yokohama 236-0004, Japan; smaeda@yokohama-cu.ac.jp

**Keywords:** detective flow imaging, pancreatic cancer, endoscopic ultrasonography

## Abstract

The differential diagnosis of solid pancreatic lesions (SPLs) using B-mode endoscopic ultrasonography (EUS) is challenging. Detective flow imaging (DFI) offers the potential for detecting low-flow vessels in the pancreas, thus enhancing diagnostic accuracy. This retrospective study aimed to investigate DFI-EUS findings of SPLs and analyze their differential diagnostic accuracy for pancreatic cancer. We included 104 patients with pathologically confirmed SPLs who underwent EUS between April 2021 and June 2023. Expert endosonographers, blinded to the patients’ clinical data, evaluated images obtained through B-mode, eFLOW, and DFI-EUS. The frame rate and vessel detection sensitivity were compared between eFLOW and DFI, and the diagnostic criteria for pancreatic cancer were established. The visualization rate for vessels in SPLs was significantly higher with DFI-EUS (96%) compared to eFLOW (27%). Additionally, DFI showed a superior frame rate, sensitivity (99%), and accuracy (88%) for detecting pancreatic cancer, although with a modest specificity (43%). On DFI-EUS, characteristics such as hypovascularity, peritumoral vessel distribution, or spotty vessel form were suggestive of pancreatic cancer. DFI-EUS significantly improved the visualization of vascular structures within the SPLs, highlighting its efficacy as a diagnostic modality for pancreatic cancer.

## 1. Introduction

Ultrasonography (US) plays a crucial role in the detection of solid pancreatic lesions (SPLs) [1,2,3,4]. The identification of small hypoechoic masses or the dilation of the main pancreatic duct is the key to the early diagnosis of pancreatic cancer [5,6]; however, the artifacts generated by abdominal gas and ultrasound attenuation owing to the distance from the body surface have limited the diagnostic application of US. Endoscopic ultrasonography (EUS) is integral to the differential diagnosis and screening of SPLs [7,8,9] owing to its superior spatial resolution and reduced abdominal gas interference. Nevertheless, differentiating various types of SPLs based on the findings of conventional EUS is challenging.

Most lesions are visualized as hypoechoic masses in B-mode EUS, and the dilation of the main pancreatic duct is insufficient for diagnosing pancreatic cancer [10]. Contrast-enhanced computed tomography (CT) is used for the diagnosis of SPLs [2,11]; however, EUS has not been applied for the vascular evaluation of SPLs. Although Doppler imaging provides information regarding the major vessels surrounding the pancreas [12], the details of minute vessels within the SPLs cannot be obtained via this modality [13].

Contrast-enhanced EUS (CE-EUS) is used for the differential diagnosis of SPLs [11,14,15,16,17,18,19,20]. However, some pancreatic cancer lesions are misdiagnosed as mass-forming pancreatitis (MFP) as they appear iso- or hypervascular on CE-EUS owing to its high sensitivity. An evaluation of the “wash-out sign” [21] has been recommended to mitigate this issue; nevertheless, these limitations of CE-EUS hinder its usage.

Recently, a novel technology using US, such as superb microvascular imaging (SMI) (Canon Medical System Corporation, Tokyo, Japan), B-flow imaging (GE Healthcare, Milwaukee, WI, US), and detective flow imaging (DFI) (Fujifilm Healthcare, Tokyo, Japan) were developed for the detection of slow-flow vessels (Figure 1) [22,23,24,25,26,27,28]. These modalities, which are characterized by a high frame rate, have enabled the visualization of minute tumor vessels while minimizing the incidence of motion artifacts. DFI-EUS is sensitive to the detection of vessels; however, DFI has not been applied for the differential diagnosis of SPLs [29,30,31].

This study aimed to investigate DFI-EUS findings of SPLs and analyze their differential diagnostic accuracy for pancreatic cancer.

## 2. Materials and Methods

### 2.1. Patients

Medical records of 104 patients with pathologically confirmed SPLs who had undergone EUS examinations between April 2021 and June 2023 were retrospectively analyzed in this study. The inclusion criteria required participants to be over 18 years old, with archived B-mode, eFLOW, and DFI images. Pathological findings were acquired via surgery, EUS-guided fine needle biopsy (EUS-FNB), and fluoroscopic biopsy using endoscopic retrograde cholangiopancreatography. A diagnosis of MFP was established after confirming the absence of malignant findings both on EUS-FNB and over 6 months follow-up with imaging modalities other than EUS. This study was approved by the Ethics Committee of Yokohama City University (F220900020) and adhered to the ethical standards outlined by the Institutional Research Committee and the latest version of the Declaration of Helsinki. As this study was retrospective, obtaining patient consent was not required, and information was shared through an opt-out process.

### 2.2. EUS Examination

All endoscopic examinations were performed or supervised by experienced endosonographers. EUS was performed using ARIETTA 850 (FUJIFILM Healthcare, Tokyo, Japan) and GF-UCT260 (Olympus Medical Systems, Tokyo, Japan). Propofol, midazolam, and/or diazepam were administered to induce sedation. The doses were adjusted according to the body weights and ages of the patients. The SPLs, including the surrounding pancreatic parenchyma, were scanned using B-mode to acquire high-quality ultrasound images of the scan region. eFocusing was implemented during the acquisition. eFLOW was set to approximately 5.0 cm/s in the velocity range. Subsequently, DFI was used to evaluate the vascular structure within and surrounding the SPLs. The optimal color gain was adjusted based on the size and depth of the SPLs. A region of interest (ROI), including the SPLs and adjacent pancreatic parenchyma, was defined for each SPL. All images were stored in a digital format on an ARIETTA 850 hard disk by each endosonographer. In addition, static vessel images of the largest plane of each SPL were acquired for blinded reading.

### 2.3. Image Evaluation

The long diameter, depth of the SPLs, and frame rates for eFLOW and DFI were determined from the static images. All B-mode, eFLOW, and DFI images were numbered. Three expert endosonographers, each with over 5 years of experience in EUS and 1 year of experience in DFI-EUS, evaluated the images. Clinical information and final diagnoses were blinded. Figure 1 presents the evaluation criteria. The images were classified as fair or poor regarding evaluability. Echogenicity (hyperechoic, isoechoic, or hypoechoic), border detection (well-defined or indistinct), contour shape (smooth or irregular), and the internal echo of the SPLs (homogeneous or heterogeneous) were classified on B-mode images. The presence of vessels (present or absent), vessel distribution (peritumoral or intratumoral), and vessel form (spotty or linear) in the SPLs was classified on eFLOW images. The presence of vessels (present or absent), vascularity (hypervascular or hypovascular), vessel distribution (peritumoral or intratumoral), and vessel shape (spotted or linear) in the SPLs were classified on the images acquired via DFI (Figure 2). The eFLOW and DFI findings were evaluated only in cases where vessels were present within the SPLs. The “vascularity” on DFI was classified as hypovascular in cases where vessels were not present within the SPLs. When a discrepancy was observed in the findings made by the three readers, the findings selected by the majority of the two were adopted. Diagnostic criteria for pancreatic cancer were formulated based on these findings. The findings of lesions other than pancreatic cancer were also evaluated. Neuroendocrine neoplasm (NEN) and metastasis from renal cell carcinoma (RCC) were collectively classified as “hypervascular tumors” based on historical results [32,33,34].

### 2.4. Study Design

This was a retrospective study that was designed to evaluate the usefulness of DFI-EUS for SPL diagnosis. The primary outcome was the accuracy of the differential diagnosis for pancreatic cancer. The secondary outcomes were the frame rates and vessel detection rates of eFLOW and DFI. The relevant outcome was the differential diagnosis between MFP and NEN.

### 2.5. Statistical Analysis

All statistical analyses were performed using the JMP Pro 17 (SAS Institute Inc., Cary, NC, USA). Categorical variables are presented as frequencies with percentages. Continuous variables are presented as medians and ranges. The frame rate and vessel detection rate of eFLOW and DFI were compared using Student’s *t*-test and McNemar’s test. Categorical variables for the characteristics and US findings of pancreatic cancer and those of other lesions were compared using the chi-square. Statistical significance was set at *p*-value < 0.05. Diagnostic criteria for pancreatic cancer were established for each modality based on the significant findings identified via Fisher’s exact test. The sensitivity, specificity, positive predictive value (PPV), negative predictive value (NPV), and accuracy were determined based on the diagnostic criteria for pancreatic cancer.

## 3. Results

### 3.1. Patients’ Characteristics

Table 1 presents the patient characteristics. Among the 104 patients included in this study, 79 (76%), 9 (8.7%), 6 (5.8%), 3 (2.9%), 2 (1.9%), and 1 (1.0%) were diagnosed with pancreatic cancer, MFP, NEN, pancreatic metastasis from RCC, intrapancreatic accessory spleen (IPAS), and malignant lymphoma, respectively. The lesions were located in the pancreatic head and body or tail in 50 (48%) and 54 (52%) patients, respectively. Fifty-six (54%) patients underwent transgastric scanning. The median lesion diameter of lesions was 21 mm (range: 6–53 mm). The depth of the lesions (distance from the echoendoscope to the bottom of the tumor) was 27.5 mm (range: 13–53 mm).

### 3.2. Frame Rates of eFLOW and DFI

The frame rate was set as 20 frames per second (fps) for B-mode; in contrast, the frame rate was automatically modified according to the size of the ROI for eFLOW and DFI. Each frame rate was recorded and analyzed retrospectively (Figure 3). The median (range) of frame rate for DFI was significantly higher than that for eFLOW (43 [31–96] fps vs. 12 [8–15] fps; *p* < 0.01).

### 3.3. Vessel Detection Rates of eFLOW and DFI

DFI-EUS visualized vessels within the SPLs in 96% (100) of cases, whereas eFLOW achieved this in only 27% (28) of cases. Thus, the vessel detection sensitivity of DFI-EUS was significantly higher than that of eFLOW (*p* < 0.01).

### 3.4. Findings of Pancreatic Cancer

Table 2 presents the result of the univariate analysis of the characteristics and findings of pancreatic cancer and other lesions. Pancreatic cancer was observed significantly more frequently in the pancreatic head (54%, *p* = 0.013) and in areas deeper than 25 mm (63%, *p* = 0.042). The number of large SPLs (>20 mm) was greater in patients with pancreatic cancer than in other types of pancreatic lesions; however, the difference was not statistically significant. The majority of lesions were classified as hypoechoic lesions on B-mode (98% (102/104), *p* = 0.473). The borders were “well-defined” in 70% of the pancreatic cancer lesions and 81% of other types of lesions (*p* = 0.362). “Irregular contour” and “heterogenic internal echo” were observed in 88% (73/83) and 83% (69/83) of pancreatic cancer lesions, respectively (*p* < 0.01). eFLOW detected the presence of vessels in 27% (28/104) of lesions only. The shape and distribution of the vessels were classified in cases where vessels were present within the lesion; however, no significant differences were observed. DFI detected the presence of vessels in 96% (100/104) of the lesions. Thus, the detection rate of DFI was significantly higher than that of eFLOW (*p* < 0.01). Four lesions without vessels were classified as “hypovascular”. Hypovascular lesions were significantly more frequent in patients with pancreatic cancer (*p* < 0.01). The distribution and shape of the vessels were evaluated in 100 cases wherein the vessels were present within the lesions. Peritumoral and spotty vessels were significantly more frequent in pancreatic cancer lesions (84% [66/79] and 86% [68/79], respectively; Figure 4a) than in other types of lesions (38% [8/21] and 43% [9/21], respectively; *p* < 0.01).

### 3.5. Diagnostic Criteria for Pancreatic Cancer

The diagnostic criteria for pancreatic cancer were set as “irregular or heterogeneous” and “hypovascular or with peritumoral or spotty vessels” on B-mode and DFI, respectively. Table 2 presents the number of cases satisfying each criterion. Table 3 presents the diagnostic accuracy. Although no significant differences were observed, the “absence of vessels” on eFLOW was set as a finding of pancreatic cancer. The sensitivity, specificity, and accuracy of these diagnostic criteria for pancreatic cancer were 94%, 19%, and 44% on B-mode; 76%, 38%, and 68% on eFLOW; and 99%, 43%, and 88% on DFI, respectively.

### 3.6. Findings on Lesions Other Than Pancreatic Cancer

Table 4 presents the findings in the NEN (*n* = 6) and MFP (*n* = 9) groups. No significant differences were observed between the two groups in terms of lesion characteristics. All lesions were hypoechoic on B-mode. No significant differences were observed between the B-mode and eFLOW findings. DFI could visualize vessels within the SPLs in all lesions. Notably, 83% (5/6) of the NEN lesions were hypervascular, whereas all MFP lesions were hypovascular (Figure 4b), with a statistically significant difference (*p* < 0.01).

## 4. Discussion

To the best of our knowledge, this study is the first to demonstrate the utility of DFI-EUS for the differential diagnosis of SPLs. The ability of DFI-EUS to detect vessels within the lesions was significantly higher than that of eFLOW. Pancreatic cancer lesions are characterized by the presence of hypovascular and spotty vessels in the peritumoral region. The sensitivity, specificity, and accuracy of DFI-EUS for the detection of pancreatic cancer were 99%, 43%, and 88%, respectively, which were higher than those of B-mode and eFLOW.

Ultrasound techniques for detecting slow-velocity vessels are unique. Lu R et al. and Bakdik S et al. have reported that SMI was used for the differential diagnosis of thyroid nodules and breast cancer [22,26]. The precision of DFI for the identification of vessels is similar to that of SMI; however, only a few studies have been conducted in this field. A novel algorithm that can eliminate motion artifacts from the feature amount of movement derived by evaluating the received signals in the ROI, which vary according to the signal intensity, was used in DFI [29]. The frame rate used in the DFI setting at 43 fps (range: 31–96 fps) was higher than that of eFLOW in this study.

The ability of eFLOW and DFI to detect the presence of vessels within SPLs was compared in this study. eFLOW facilitates imaging with high spatial resolution without the occurrence of the blooming artifacts in Color Doppler imaging [12,13,35]. However, the detection of vessels within SPLs with eFLOW remains challenging as most pancreatic cancer lesions are hypovascular. A dense fibrotic stroma is observed in pancreatic cancer lesions. Thus, the blood flow within the lesion is limited compared with that in the surrounding pancreatic parenchyma [36,37,38]. The vessel detection rate of eFLOW was only 27% in this study, which is lower than that reported in previous studies. This finding may be attributed to the large proportion of pancreatic cancer lesions and the relatively small size of the hypervascular lesions in the study cohort. The vessel detection rate of DFI was significantly higher (96%) than that of eFLOW. The sensitivity of DFI for the detection of vessels has been reported in several studies; however, data related to SPLs are absent. This study’s findings may facilitate the adoption of a new approach for assessing the structure of the vessels within SPLs using DFI-EUS.

Criteria for the classification of vessels within SPLs remain unestablished. In DFI images, heterogeneity, which is useful in CE-EUS [14,21], cannot be evaluated. Thus, the distribution and shape of the vessels within SPLs were evaluated in this study. The distribution of the vessels within the SPLs was classified as “peritumoral” or “intratumoral”, based on whether the vessels were detected throughout the SPLs. The shape of the vessels was classified as “spotty” or “linear”, based on the continuity of the vessel signals. The diagnostic criteria for pancreatic cancer were set to maximize the sensitivity of B-mode, eFLOW, and DFI; thus, the specificity was relatively low. The lower specificity of these criteria may be attributed to the MFP findings. MFP lesions presented with irregular margins and heterogeneity on B-mode and hypovascular with spotty vessels on DFI. These results indicate that DFI can be used to differentiate pancreatic cancer from NEN. However, further studies are warranted to differentiate pancreatic cancer from MFP.

Although DFI-EUS offers enhanced differential diagnostic capabilities, tissue sampling with EUS-FNB remains a critical issue. Although advances in needle design have been reported recently, the accuracy of EUS-FNB remains approximately 90% in small SPLs [39]. In other cases, wherein false diagnoses occur, it becomes necessary to decide whether to re-perform EUS-FNB or follow-up with imaging modalities. Therefore, the improvement of the differential diagnosis of SPLs with DFI-EUS could improve decision making. Furthermore, DFI-EUS allows for a simplified assessment of vascularity within SPLs and can also be employed to determine the technique to use in EUS-FNB. Previous studies have shown that compared with the slow-pull method, the wet suction technique results in a higher frequency of blood contamination [40]. Thus, the identification of hypervascular lesions, such as NEN or RCC metastases, can help in the selection of the slow-pull method.

The findings of this study demonstrate the utility of DFI for purposes other than the differential diagnosis of SPLs. DFI enabled the identification of small hypervascular lesions within 10 mm, which was unexpected (Figure 4c,d). The frame rate of DFI is higher than those of B-mode and eFLOW; thus, DFI can be used as a screening evaluation for patients with Multiple Endocrine Neoplasia type 1 or von Hippel–Lindau disease [41,42]. Hypersensitivity to vascular structures aided the avoidance of thick vessels during the EUS-FNB procedure (Figure 5). DFI, rather than eFLOW, can be used to define the vessels more precisely in patients with NEN. This technique enables the effective acquisition of tissue samples without major bleeding. The absence of linear vessels in pancreatic cancer lesions can aid the detection of tumor margins (Figure 6). A pancreatic cancer lesion without a detectable margin was observed on B-mode and eFLOW in this case, and the initial EUS-FNB result was a false negative. Subsequently, DFI was performed, and the normal vessels in the surrounding pancreatic parenchyma near the tumor margin disappeared. The diagnosis was confirmed by puncturing the areas without vessels on DFI. Thus, DFI can be considered a novel tool for determining the differential diagnosis of pancreatic cancer in the future.

This study has some limitations. First, this was a single-center retrospective study with a small sample size; particularly, the number of patients with the benign disease was inadequate. Differences in the epidemiological frequencies of pancreatic cancer and other benign lesions may have affected the results of the statistical analysis. Therefore, further prospective studies must be conducted in the future to establish the diagnostic criteria of pancreatic cancer using DFI-EUS. Second, the images evaluated in this study were static vessel images with the largest planes of the SPLs. The vessel distribution was not homogeneous; thus, the results could differ in other planes. Third, the final diagnosis of the SPLs was not confirmed surgically or through FNB. Although benign lesions can be detected using other imaging modalities, misdiagnosis can alter the diagnostic criteria. Finally, DFI and CE-EUS findings were not compared in this study because the use of contrast agents for SPLs has not been approved under the Japanese health insurance system. DFI has the advantage of not requiring additional materials; however, further prospective studies are needed to compare the accuracy of CE-EUS.

## 5. Conclusions

In conclusion, DFI-EUS is a superior modality for the detection of pancreatic cancer as it significantly enhances the visualization of vascular structures within the SPLs. This novel vascular imaging modality, which does not require a contrast agent, is more useful for differential diagnosis than B-mode or eFLOW.

## Figures and Tables

**Figure 1 diagnostics-14-00882-f001:**
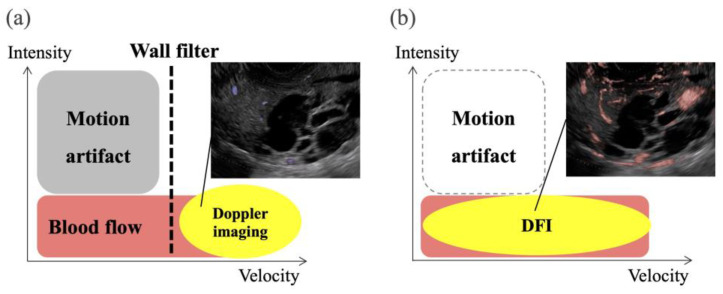
Schema of eFLOW and detective flow imaging (DFI). (**a**) eFLOW reduces motion artifacts and low-flow vessels using a wall filter. (**b**) DFI enables the detection of low-flow vessels while reducing motion artifacts.

**Figure 2 diagnostics-14-00882-f002:**
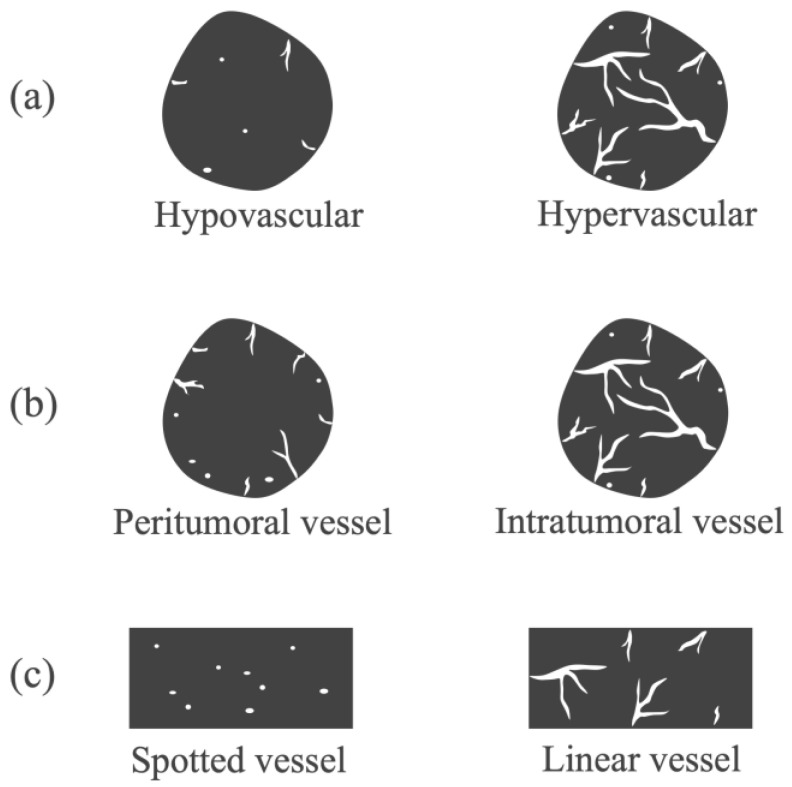
Diagnostic criteria for DFI. (**a**) Vascularity was classified as hypovascular or hypervascular. (**b**) Vessel distribution was classified as peritumoral or intratumoral. (**c**) Vessel shape was classified as spotted or linear.

**Figure 3 diagnostics-14-00882-f003:**
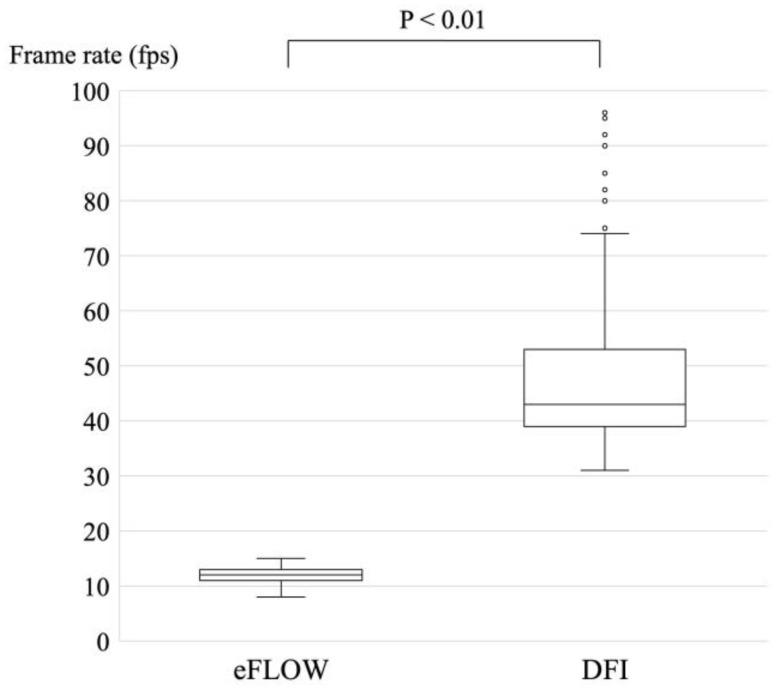
Comparison of the frame rate between eFLOW and DFI. The median frame rate for DFI (43 fps) was significantly higher than that of eFLOW (12 fps).

**Figure 4 diagnostics-14-00882-f004:**
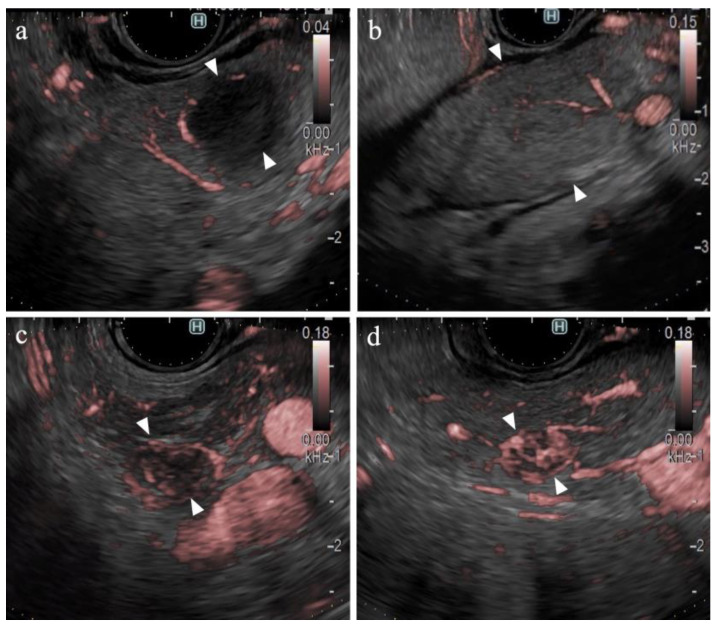
Vessel images of DFI. (**a**) Pancreatic cancer; (**b**) mass-forming pancreatitis; (**c**) neuroendocrine neoplasm; and (**d**) metastatic tumor (renal cell carcinoma). The arrowheads in each figure show the margin of the SPLs.

**Figure 5 diagnostics-14-00882-f005:**
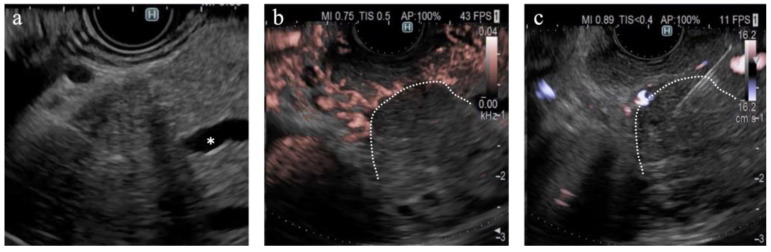
A case of pancreatic cancer. (**a**) On B-mode, the main pancreatic duct is obstructed in the pancreatic head (*); however, the tumor margin is unclear. (**b**) On DFI, the tumor margin is detectable because of the interruption of vessels in the pancreatic parenchyma. (**c**) Endoscopic ultrasound-guided tissue acquisition was successfully performed according to the margin on DFI.

**Figure 6 diagnostics-14-00882-f006:**
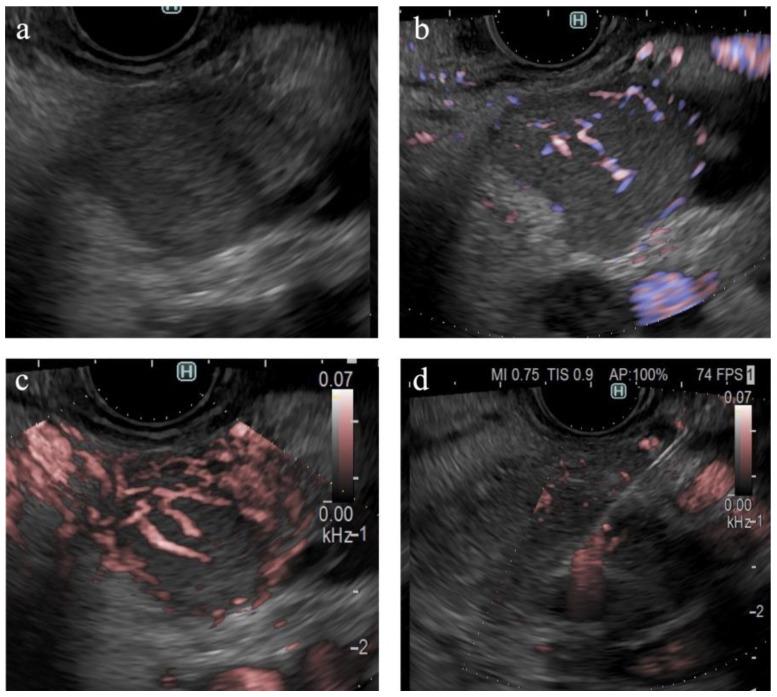
A case of pancreatic neuroendocrine neoplasm. (**a**) B-mode shows a hypoechoic tumor with a well-defined margin. (**b**) eFLOW shows linear vessels in the lesion. (**c**) On DFI, the tumor is occupied by dilated vessels. (**d**) Endoscopic ultrasound-guided tissue acquisition.

**Table 1 diagnostics-14-00882-t001:** Patient characteristics.

	N = 104
Age, median (range), years	72.5 years (38–86)
Sex, male, n (%)	54 (52%)
Disease, n (%)	
Pancreatic cancer	83 (80%)
Mass-forming pancreatitis	9 (8.7%)
Neuroendocrine neoplasm	6 (5.8%)
Pancreatic metastasis from RCC	3 (2.9%)
Intrapancreatic accessory spleen	2 (1.9%)
Malignant lymphoma	1 (1.0%)
Location of lesions, n (%)	
Pancreatic head	50 (48%)
Pancreatic body/tail	54 (52%)
Transgastric scan, n (%)	56 (54%)
Diameter of lesions, median (range)	21 mm (6–53)
Depth of lesions, median (range)	27.5 mm (13–53)

RCC: renal cell carcinoma.

**Table 2 diagnostics-14-00882-t002:** Findings of pancreatic cancer.

Modality	Category	Findings	Pancreatic Cancer *n* = 83	Other Lesions *n* = 21	*p* Value
Characteristics	Age	>75 years	31/83, 37%	5/21, 24%	0.244
	Sex	Male	42/83, 51%	13/21, 62%	0.354
	Location	Pancreatic head	45/83, 54%	5/21, 24%	0.013
	Transgastric scan		41/83, 49%	15/21, 71%	0.070
	Diameter of lesions	>20 mm	50/83, 60%	8/21, 38%	0.068
	Depth	>25 mm	52/83, 63%	8/21, 38%	0.042
B-mode	Echogenicity	Hypoechoic	81/83, 98%	21/21, 100%	0.473
	Border	Indistinct	24/83, 30%	4/21, 19%	0.362
	Contour	Irregular	73/83, 88%	10/21, 48%	<0.01
	Internal echo	Heterogeneous	69/83, 83%	11/21, 52%	<0.01
	Diagnostic criteria for pancreatic cancer	Irregular or Heterogeneous	78/83, 94%	17/21, 81%	0.058
eFLOW	Vessel detection	Present	20/83, 24%	8/21, 38%	0.196
	Vessel distribution	Peri-tumoral	17/20, 85%	5/8, 63%	0.200
	Vessel shape	Spotty	14/20, 70%	4/8, 50%	0.318
DFI	Vessel detection	Present	79/83, 95%	21/21, 100%	0.305
	Vascularity	Hypovascular	80/83, 96%	11/21, 52%	<0.01
	Vessel distribution	Peritumoral	66/79, 84%	8/21, 38%	<0.01
	Vessel shape	Spotty	68/79, 86%	9/21, 43%	<0.01
	Diagnostic criteria for pancreatic cancer	Hypovascular or Peritumoral or Spotty	82/83, 99%	12/21, 57%	<0.01

DFI: detective flow imaging.

**Table 3 diagnostics-14-00882-t003:** Diagnostic accuracy for pancreatic cancer.

Modality	Findings	Sensitivity	Specificity	PPV	NPV	Accuracy
B-mode	Irregular or Heterogeneous	94% (78/83)	19% (4/21)	82% (78/95)	44% (4/9)	79% (82/104)
eFLOW	Absence of vessels	76% (63/83)	38% (8/21)	83% (63/76)	29% (8/28)	68% (71/104)
DFI	Hypovascular or Peritumoral or Spotty	99% (82/83)	43% (9/21)	87% (82/94)	90% (9/10)	88% (91/94)

DFI: detective flow imaging, PPV: positive predictive value, NPV: negative predictive value.

**Table 4 diagnostics-14-00882-t004:** Findings of lesions other than pancreatic cancer.

Modality	Category	Findings	NEN *n* = 6	MFP *n* = 9	*p* Value
Characteristics	Age	>75 years	1/6, 17%	2/9, 22%	0.792
	Sex	Male	3/6, 50%	7/2, 78%	0.264
	Location	Pancreatic head	1/5, 17%	3/9, 33%	0.475
	Transgastric scan		4/6, 67%	6/9, 67%	1.000
	Diameter of lesions	>20 mm	1/6, 17%	5/9, 56%	0.132
	Depth	>25 mm	1/6, 17%	5/9, 56%	0.132
B-mode	Echogenicity	Hypoechoic	6/6, 100%	9/9, 100%	-
	Border	Indistinct	0/6, 0%	4/9, 44%	0.057
	Contour	Irregular	2/6, 33%	7/9, 78%	0.085
	Internal echo	Heterogeneous	2/6, 33%	7/9, 78%	0.085
eFLOW	Vessel detection	Present	2/6, 33%	2/9, 22%	0.634
DFI	Vessel detection	Present	6/6, 100%	9/9, 100%	-
	Vascularity	Hypovascular	1/6, 17%	9/9, 100%	< 0.01
	Vessel distribution	Peritumoral	1/6, 17%	5/9, 56%	0.132
	Vessel shape	Spotty	1/6, 17%	6/9, 67%	0.572

NEN: neuroendocrine neoplasm, MFP: mass-forming pancreatitis, DFI: detective flow imaging.

## Data Availability

The datasets generated during and/or analyzed during this study are available from the corresponding author upon reasonable request.

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
