# Peer review of "Differential Diagnosis of Solid Pancreatic Lesions Using Detective Flow Imaging Endoscopic Ultrasonography"

_diagnostics, 2024, doi:10.3390/diagnostics14090882_

Round 1
Reviewer 1 Report
Comments and Suggestions for Authors
This is a very interesting study on a novel technology called detective flow imaging (DFI).
1) Please, double check Figure 2 because to seems not related to the legend.
2) Please, state clearly in the method section the primary and secondary aims, and the relative outcomes.
3) I don't see figure 3
4) How was the final diagnoses made? Please specify, especially because mass forming pancreatitis may be challanging
5) There is no comparison with CH-EUS. This is a major limitation of this study that must be mentioned and discussed.
6) EUS-FNB is still needed for the diagnosis of SPL. Please mention this point in the discussion by citing PMID: 35915956
Reviewer 2 Report
Comments and Suggestions for Authors
A very interesting study by Miwa et al. that evaluates the role of flow imaging endoscopic US in the differential diagnosis of pancreatic lesions. The study is novel and the research question of significant clinical importance. The manuscript is well written and appropriately structured. One minor comment:
-Besides chi-square test in the various endpoints, wouldn't it be appropriate to, also, estimate the respective AUCs and compare them?
Comments on the Quality of English LanguageMinor linguistic and phraseological revisions from a professional native speaker should be considered.
Round 2
Reviewer 1 Report
Comments and Suggestions for Authors
I have no further comments